# Characterization and Preliminary Application of a Novel Lytic *Vibrio parahaemolyticus* Bacteriophage vB_VpaP_SJSY21

**DOI:** 10.3390/ijms242417529

**Published:** 2023-12-15

**Authors:** Yajin Xu, Jun Sun, Jingjie Hu, Zhenmin Bao, Mengqiang Wang

**Affiliations:** 1Southern Marine Science and Engineering Guangdong Laboratory (Guangzhou), Guangzhou 511458, China; xuyajin@stu.ouc.edu.cn (Y.X.); sunjun@stu.ouc.edu.cn (J.S.); zmbao@ouc.edu.cn (Z.B.); 2MOE Key Laboratory of Marine Genetics and Breeding, College of Marine Life Sciences, Ocean University of China, Qingdao 266003, China; 3Key Laboratory of Tropical Aquatic Germplasm of Hainan Province, Sanya Oceanographic Institution, Ocean University of China, Sanya 572040, China

**Keywords:** bacteriophage, *Vibrio parahaemolyticus*, *Litopenaeus vannamei*, whole-genome sequencing, gut microbiota

## Abstract

*Litopenaeus vannamei* is one of the most economically significant aquatic species globally. However, the emergence of acute hepatopancreatic necrosis disease (AHPND) in recent years has resulted in substantial losses within the *L. vannamei* farming industry. Phage therapy holds promise as an effective strategy for preventing and controlling bacterial infections like AHPND, thereby promoting the healthy and sustainable growth of the shrimp aquaculture sector. In this study, a novel and unique *Vibrio parahaemolyticus* bacteriophage, named vB_VpaP_SJSY21, was successfully isolated from sewage samples. Using transmission electron microscopy, it was observed that phage SJSY21 has an elongated shell. Notably, phage SJSY21 exhibited high infection efficiency, with an optimal multiplicity of infection (MOI) of only 0.01 and a remarkably short latent period of 10 min, resulting in a lysis quantity of 508. Furthermore, phage SJSY21 demonstrated notable heat resistance and the capacity to withstand high temperatures during preservation, thus holding potential for application in phage therapy. Whole-genome sequencing and analysis confirmed that phage SJSY21 has a genome size of 110,776 bp, classifying it as a new member of the short-tailed bacteriophage family. Additionally, cultivation experiments indicated that phage SJSY21 has the potential to enhance the survival of *L. vannamei* in culture systems, thereby offering innovative prospects for the application of phage therapy in aquaculture.

## 1. Introduction

The shrimp farming industry assumes a pivotal role in global aquatic food supply. Nowadays, *Litopenaeus vannamei* stands as one of the foremost seafood products worldwide, surpassing the combined production of other shrimp species [1]. However, the intensification of *L. vannamei* farming has precipitated an escalating problem of diseases. Bacterial diseases represent a major challenge in shrimp farming, and acute hepatopancreatic necrosis disease (AHPND) has emerged as a recent and profoundly impactful ailment [2]. Previous reports have identified *Vibrio parahaemolyticus* strains carrying pVA1 plasmids containing toxin genes pirA and pirB as pivotal pathogens of AHPND [3,4]. AHPND is characterized by its high infectivity and mortality rates, causing severe damage to the digestive glands, such as the hepatopancreas and intestines. This detrimentally affects their feeding capacity, leading to a substantial incidence of sudden deaths during early growth stages [5,6]. At present, AHPND causes significant production losses and other negative effects in the global shrimp farming industry, with estimated annual losses exceeding USD 1 billion [7,8].

The classic approach to the defense and treatment of bacterial diseases in aquaculture involves the addition of antibiotics and other antibacterial agents. Commonly used medications include amides, tetracyclines, and sulfonamides. However, these agents often exhibit broad-spectrum bactericidal effects, disrupting the shrimp’s gut microbiota equilibrium and increasing the risk of colonization by other pathogenic organisms. Moreover, their administration in aquaculture, whether in feed or water, leads to substantial drug residues, causing ecological damage to the surrounding environment and potentially impacting human health. The issue of antibiotic misuse has exacerbated the problem of antibiotic resistance. Reports indicate that pathogenic strains of Mexico’s *V. parahaemolyticus* (13-511/A1 and 13-306D/4) carry the tetB gene encoding tetracycline resistance [2], while China’s *V. campbellii* harbors multiple antibiotic resistance genes [9]. Consequently, the use of antibiotics and similar drugs in shrimp farming may not provide a sustainable control measure against AHPND, leading to the gradual abandonment of this strategy [10].

Bacteriophages are viruses that can infect bacteria and replicate within bacteria. Phage therapy has demonstrated substantial potential application value in the treatment and prevention of animal diseases [11]. Phages have been employed in countering infections caused by *Escherichia coli*, Salmonella, and *Staphylococcus aureus* in cattle, pigs, and chickens. In the realm of aquaculture, antibiotics are gradually being prohibited due to the mounting issue of antibiotic resistance. However, phage therapy to treat AHPND in *L. vannamei* farming has the obvious advantages of high specificity, stability, and safety. Research on the *V. parahaemolyticus* phage remains limited, with a dearth of mature application-oriented research [12].

In this study, the biological and genomic properties of a newly isolated *V. parahaemolyticus* phage, vB_VpaP_SJSY21, were determined and its effect on the intestinal microbial community of *L. vannamei* was analyzed in order to (1) isolate and characterize this novel *V. parahaemolyticus* phage SJSY21, (2) explore the preventive effect of phage SJSY21 in the culture of *L. vannamei*, and (3) develop a safer treatment method for targeted elimination of *Vibrio* controlling AHPND and provide new methods for the further study of phage therapy in the aquatic field.

## 2. Results

### 2.1. Isolation, Purification, and General Characterization of Phages

The isolation experimental results showed that this new type of bacteriophage had the ability of infecting the host bacteria *V. parahaemolyticus*, and was therefore named vB_VpaP_SJSY21. The initial phage formed a uniform plaque with neat edges after three rounds of double-layer plate purification. The morphological characteristics of phage SJSY21 observed using TEM are shown in Figure 1. The SJSY21 phage was similar to the short-tailed bacteriophage family members, with a head length of 190 nm and a total length of 220 nm.

### 2.2. MOI and One-Step Growth

As shown in Figure 2, phage SJSY21 has the highest titer at a MOI of 0.01, which means that the optimal MOI is 0.01, and its titer is 4.9 × 10^9^ PFU/mL (*p* < 0.05). As shown in Figure 3, the incubation period of this bacteriophage was 10 min, the lysis period was 50 min, and the lysis amount was 508.

### 2.3. Results of Lysis and Temperature Stability Assay

The result of the lysis assay is shown in Figure 4. It can be seen that no bacteriophage spots were observed in A–D, and only obvious bacteriophage spots were observed in E, indicating that phage SJSY21 did not interact with the four probiotics selected in this study. As shown in Figure 5, phage SJSY21 maintained high activity at 40 °C for 90 min, while at 50 °C and above the titer of this bacteriophage decreased within 90 min. After treatment, at the highest temperature of 80 °C for 90 min, the titer was 2.6 × 10^2^ PFU/mL.

### 2.4. Characteristics of Phage Genome

The genome of phage SJSY21 can only be digested by DNase I, indicating that the genomic nucleic acid type of phage SJSY21 is dsDNA. A total of 20.38 Mb of raw data reads with a data volume of 3.07 Gb was obtained after whole-genome sequencing, and the results are shown in Table 1. The effective reads obtained were 20.29 Mb with a data volume of 3.04 Gb. The genome of phage SJSY21 is linear dsDNA with a total length of 110,776 bp, and the statistics of the assembly results are shown in Table 2. The analysis results of blast n showed that phage SJSY21 has a genetic relationship with two strains of bacteriophages, with a coverage rate of 77% with *V. parahaemolyticus* bacteriophage Vp_R1 (MG603697.1) and 2% with *V. alginolyticus* bacteriophage vB_ValP_VA-RY-3 (MW900438.1). The annotation results obtained with GeneMarkS showed that the genome of phage SJSY21 contained a total of 196 open read frames (ORFs), with a total length of 102390 bp. On average, there were 1.77 genes per 1 kb base, with a maximum ORF length of 6624 bp. The total length of all ORFs accounted for 92.43% of the total genome length, and the average GC content of the ORFs was 41.16%.

### 2.5. Protein Function Prediction of Bacteriophages

A total of 138 ORFs were annotated in phage SJSY21, accounting for 70.41% of the total number of ORFs, shown in Appendix A, while a total of 19 ORFs were annotated using the eggNOG database, accounting for 9.69% of the total ORFs. The annotated results are shown in Table 3. A total of seven pathways were annotated, and the statistical results of the KEGG pathway are shown in Figure 6A. InterPro was utilized to enrich the GO terms of phage SJSY21, with a total of 12 GO terms enriched for this phage. The annotated results are shown in Figure 6B. Then, a complete genome circle diagram of the bacteriophage was drawn based on the sequence and annotation information of the genome of phage SJSY21, which is shown in Figure 7.

### 2.6. Analysis of α Diversity and β Diversity

As shown in Figure 8, the Shannon index and Simpson index of the samples in the α diversity analysis were calculated and box plots were plotted. The Shannon and Simpson diversity indices are indicators used to quantify the diversity of a community, representing the richness of species within the community [13]. The results showed that the Shannon and Simpson indices of the samples of the microbial community in the phage SJSY21 group were significantly reduced compared with the control group (*p* < 0.05). The analysis of β diversity is shown in Figure 9. The points representing the phage SJSY21 and control groups of the samples are clustered into two parts, indicating differences in microbial diversity between the two groups of samples.

### 2.7. Analysis of Community Composition Structure and Differential Species

The community composition analysis results of shrimp gut samples in each group are shown in Figure 10. The dominant strains in the phage SJSY21 group are *V. harveyi* (87.62%), *Photobacterium damselae* (4.26%), *V. parahaemolyticus* (1.59%), and *Marinifilum fragile* (1.19%). The dominant strains in the control group are *V. harveyi* (84.71%), *P. damselae* (5.79%), *V. parahaemolyticus* (1.40%), and *M. fragile* (1.39%). The abundance of *V. harveyi* in the phage SJSY21 group was significantly higher than that in the control group (*p* < 0.05). Compared with the control group, there was a change in the gut microbiota of shrimp in the phage SJSY21 group. The results of the differential species analysis are represented by a heatmap in Figure 11. The analysis results showed that the two groups ranked in the top 20 species in terms of abundance significance at the species level.

## 3. Discussion

Phages are widely distributed in nature and represent the most abundant organisms on earth, estimated at a total of 10^31^ [14]. Phages typically thrive in the ecosystems cohabited by their host bacteria. Sewage from the aquatic market with similar salinity and pH values to the survival conditions of *V. parahaemolyticus* was selected for the isolation in this study. Then, a novel *V. parahaemolyticus* bacteriophage was successfully isolated and named as vB_VpaP_SJSY21. Phage SJSY21 exhibits the typical characteristics of short-tailed phages, featuring an elongated shell and a short tail. The protein tail plays a pivotal role in facilitating phage adherence to the surface of the host bacteria [15]. The infection efficiency of phage SJSY21 is remarkably high, with an optimal infection complex of only 0.01. Phage SJSY21 demonstrates a substantial lysis capacity, reaching 508, surpassing the *V. parahaemolyticus* bacteriophage vB_VpaP_DE10 reported by Ye et al. [16]. Additionally, phage SJSY21 does not interact with several common probiotics, suggesting its potential for pathogen control by making mixtures with those probiotics. Furthermore, the results of the temperature stability assay indicate that phage SJSY21 can withstand elevated temperatures during storage and transportation.

The complete genome sequence of phage SJSY21 encodes the outer shell functional region and various enzymes necessary for DNA replication, consistent with other *V. parahaemolyticus* bacteriophages. Antibiotic resistance genes and virulence genes can be transmitted between bacteria through plasmids, transposons, bacteriophages, etc. It is well documented that bacteriophages may be a potential library for acquiring and spreading antibiotic resistance and virulence genes [17]. In this study, no toxin genes, virulence genes, antibiotic resistance genes, or integrase genes were detected in the genome of phage SJSY21. This observation underscores the safety of phage SJSY21 as a therapeutic agent against *V. parahaemolyticus* infection. Furthermore, no lysogen-related genes were identified in the phage SJSY21 genome, indicating that this bacteriophage is lytic and thus more suitable for phage therapy [18]. According to reports, the host range of bacteriophages is related to tail fibers or receptor-binding proteins, and mutations in tail-related proteins may alter the host range of bacteriophages [19]. In the genome of phage SJSY21, a tail protein encoded by ORF 165 has been predicted, which may be associated with the binding of phage SJSY21 to the host. The KEGG annotation analysis of the genome sequence of phage SJSY21 suggests that this phage genome is involved in various biological processes, including genetic information processing, metabolism, and immune response. The implication of immune system involvement suggests that phage SJSY21 may play a role in regulating the interaction between the phage and its host. In addition, the bacteriophage genome of phage SJSY21 encompasses 16 different GO categories, including genetic information processing, suggesting that phage SJSY21 may participate in regulating gene expression and controlling cellular processes.

The gut microbiota structure of *L. vannamei* plays a crucial role in maintaining the health and immune function. Therefore, the 2bRAD-M technique was employed in this study to analyze the impact of bacteriophages on the gut microbiota of shrimps [20,21]. The α diversity was significantly decreased, implying a shift in microbial community composition after the introduction of SJSY21 into the culture system. The treatment with phage SJSY21 may, to some extent, control *V. parahaemolyticus* or indirectly influence the growth of other bacterial pathogens, resulting in the enhanced health and survival of *L. vannamei* in the aquaculture system. However, previous studies have shown that dysbiosis of the gut microbiota can also increase the likelihood of shrimp disease [22]. In the samples of the phage SJSY21 group, *V. harveyi* emerged as the dominant species, constituting a substantial proportion of the total abundance, a trend similar to that in the control group. Nevertheless, the relative abundance of *V. harveyi* in the phage SJSY21 group was significantly higher than that in the control group (*p* < 0.05). This observation suggests that phage SJSY21 effectively regulates the abundance of *V. parahaemolyticus* in the gut of shrimp, thus potentially creating more growth space for *V. harveyi*. Subsequent analysis was conducted on the differential microbial species in the two groups of intestinal samples, elucidating the interactions that influence the overall microbial community structure and function. *Vibrio* genus consistently dominates the gut microbiota of shrimp, with six out of twenty different species belonging to this genus. This may be attributed to shifts in the abundance of *V. parahaemolyticus*, thereby influencing the growth of other *Vibrio* species. In addition to *Vp*_AHPND_, there exist numerous non-pathogenic *V. parahaemolyticus* strains in the gut of shrimps. While phage SJSY21 exhibits strong specificity by reducing specific *Vp*_AHPND_, the abundance of other *V. parahaemolyticus* types increases. This may explain the absence of a significant difference in the abundance of *V. parahaemolyticus* between the two groups at the species level in this study (*p* < 0.05). Some microbial species compete for the same resources, while others may facilitate the growth of additional microbial species. The abundance of specific microbial species can impact the quantities of other microbial species, thereby influencing the overall microbial community structure. These results indicate the potential of phage SJSY21 in enhancing the healthy survival of *L. vannamei* in culture systems, offering promising insights for future phage therapy applications in the aquaculture industry.

## 4. Materials and Methods

### 4.1. Isolation, Identification, and Purification of Phages

The sewage utilized for isolating the bacteriophage samples was procured from Yazhou Seafood Market in Sanya, China in 2021. Bacteriophage isolation was achieved using the double-layer agar method, with a *V. parahaemolyticus* strain serving as the host and subject of identification. A 45 mL sample of collected sewage was mixed with *V. parahaemolyticus* suspension (OD_600nm_ = 1.0) and incubated overnight at 28 °C and 160 rpm. A total of 5 mL of the incubated liquid was centrifuged at 12,000 rpm for 5 min, and the supernatant was filtered with 0.22 μm filters to obtain the bacteriophage suspension.

In this study, the Adam’s double-layer agar method was optimized to identify this novel type of bacteriophages [23]. The SM buffer was used to dilute the gradient of the phage suspension. A 100 μL dilution solution with appropriate dilution gradient, 200 μL of *V. parahaemolyticus* suspension, and 5 mL LB semi-solid medium were mixed thoroughly, then transferred at approximately 50 °C, and then the mixture was poured onto the prepared LB solid medium. This double-layer medium was incubated at 28 °C overnight to observe the presence of bacteriophage plaque. A single clear plaque was selected and the double-layer agar assay was repeated three times to ensure the purity of this novel phage. The titer of bacteriophages, also known as plaque forming unit (PFU), represents the number of infectious bacteriophages per milliliter of sample. Samples with phage titers above 10^9^ PFU/mL were selected and stored in a 4 °C for subsequent experiments.

### 4.2. Morphological Characterization

Transmission electron microscopy (TEM) was used to observe the morphology of phage SJSY21. Thirty microliters of phage suspension was drawn and dripped onto the copper mesh for one minute; the excess liquid was drawn off with filter paper. Ten microliters of phosphotungstic acid (3%, pH 7.0) was added dropwise to the copper mesh for 2 min for staining [24]. Then, the copper mesh was observed under an FEI Tecnai 12 transmission electron microscope (ThermoFisher Scientific, Waltham, MA, USA) at a voltage of 120 kV.

### 4.3. Determination of Phage Titer

In this study, the double-layer agar method was used to determine the titer of phage SJSY21. Each dilution was set to three parallels. A plate with bacteriophage plaque counts of 20-200 was considered countable. The following formula was used to calculate the titer of the bacteriophages [25]:$$\mathrm{Titer}\;\mathrm{of}\;\mathrm{bacteriophages}\;\left(\mathrm{PFU}/\mathrm{mL}\right)=\frac{\mathrm{Dilution}\;\mathrm{ratio}\times \mathrm{Average}\;\mathrm{number}\;\mathrm{of}\;\mathrm{bacteriophages}\;\mathrm{plaques}}{\mathrm{Volume}\;\mathrm{of}\;\mathrm{sample}}$$

### 4.4. Determination of Multiplicity of Infection (MOI) and One-Step Growth

The multiplicity of infection (MOI) of bacteriophages refers to the ratio of the number of bacteriophages to bacteria during infection, while the optimal MOI refers to the number of infections that could cause bacteriophages to grow at their optimal state. Five different MOIs (0.001, 0.01, 0.1, 1, 10, and 100) were set and each parallel was repeated three times. To determine the growth characteristics of the phage, a one-step growth curve of it was generated, referring to the optimal MOI, and the experimental steps were modified according to Pallavi’s method. The samples were taken at 0, 10, 20, 30, 40, 50, 60, 70, 80, 90, 100, 110, and 120 min.

### 4.5. Lysis Assay and Temperature Stability Assay

Four common probiotics in aquaculture were selected to determine whether the phage interacts with them, including *Bacillus licheniformis*, *B. subtilis* strain A, *B. subtilis* strain B, and *Clostridium butyricum*. All of them were separately mixed with gradient-diluted phage suspension to create a double-layer plate [26]. Each strain had three replicates.

To determine the temperature stability of the phage, 500 μL of phage medium with a titer of 1 × 10^9^ PFU/mL was placed at 40 °C, 50 °C, 60 °C, 70 °C, and 80 °C. The titer of each medium was determined after incubation for 30 min, 60 min, and 90 min, and each temperature had three parallels.

### 4.6. Whole-Genome Sequencing and Analysis

The DNA of the phage (titer > 10^9^ PFU/mL) was extracted by using the virus genome extraction kit (YDP315, Tiangen, Beijing, China). The obtained phage genome nucleic acid solution was treated with DNase I (RNase-free), RNase A, and Mung Bean Nuclease, and then observed with 1.5% agarose gel electrophoresis to determine its nucleic acid type. The whole genome was sequenced by Illumina NovaSeq (Illumina, San Diego, CA, USA).

The offline data were stored in FASTQ format. FastQC 0.11.8 was utilized for data quality control, followed by AdapterRemoval v2 and SOAPec v.2.0.1 to remove joint contamination and filter data. Then, A5-MiSeq and SPAdes were used to assemble the sequencing data, and MUMmer was used for collinearity analysis to obtain the final complete sequence of the phage SJSY21 genome. After obtaining the whole-genome sequence, we performed *blastn* alignment between the complete phage sequence and the NT database on NCBI. GeneMarkS was used to predict the protein-coding genes of bacterial genomes. In order to perform a functional analysis of all protein-encoded genes and phage SJSY21 at the molecular level, the protein function of the phage was predicted based on multiple databases. Diamond was used to complete the alignment of protein-coding genes. The database used for sequence alignment was the NCBI NR database. Diamond blast was used to annotate protein-coding genes with eggNOG (http://eggnog6.embl.de/), and KAAS annotation (https://www.genome.jp/tools/kaas/) and InterPro software (http://www.ebi.ac.uk/interpro/) were used to complete KEGG annotation and GO annotation analysis [15,16,17,18].

### 4.7. Experimental Animals and Immune Stimulation

The *L. vannamei* used in the current experiment were sourced from a local shrimp farm in Sanya, China. They were acclimatized in a simulated seawater environment for 7 days in a laboratory setting. The shrimps were randomly allocated into 6 separate 80 L PVC tanks with 15 shrimps per tank. The experiment consisted of two groups, each with three replicates. The treatment group received a daily addition of 10 mL of SJSY21 enrichment solution, maintaining a concentration of 1 × 10^5^ PFU/mL of bacteriophage SJSY21 in the water, while the control group received an equivalent volume of sterile water. The experiment lasted for 7 days, during which the shrimps were fed standard commercial feed at regular intervals. The culture seawater maintained a salinity of 28–30 g/L, continuous aeration, 50% daily water replacement, a temperature ranging from 28.5 to 30.0 °C, and pH ranging from 7.8 to 8.2. Then, five healthy shrimps of similar size were randomly selected from each group and their intestinal tissue was dissected and stored under sterile conditions. Five shrimps were mixed as one sample. All samples were stored at −80 °C for future experiments.

### 4.8. Extraction and Sequencing of Gut Microbiota Genome

DNA from intestines was extracted by using Cell/Tissue DNA Isolation kits (DC102-01, Vazyme, Nanjing, China). The extracted DNA was constructed and sequenced according to a previous report [20]. Two microliters of BcgI restriction endonuclease was used to digest genomic DNA at 37 °C for 3 h. Then, 0.2 μM joints (Ada1, Ada2) were used to connect and react at 4 °C for 16 h, and then heat treated at 65 °C for 20 min. The PCR amplification reaction system consisted of 7 μL connected DNA, 0.1 μM primer, 0.3 mM dNTP, 1 × HF buffer, and 0.2 μL high-fidelity DNA polymerase. The reaction program consisted of 28 cycles, including 98 °C for 5 s, 60 °C for 20 s, 72 °C for 10 s, and an extension at 72 °C for 10 min. The final database samples were purified using the QIAquick PCR purification kit (28106, Qiagen, Hilden, Germany) and subjected to 2bRAD-M high-throughput sequencing on the Nova sequencing platform.

### 4.9. Analysis of Gut Microbiota Community Structure of L. vannamei

The sequenced offline data were used for data quality control using Fastp and filtered to remove low-quality reads and obtain high-quality BcgI enzyme-digested sequence fragments. The 2bRAD-M calculation process (https://github.com/shihuang047/2bRAD-M, accessed on 1 May 2022) was used to map the Clean Reads of each sample to 173165 constructed 2bRAD-M microbial genome databases, based on the NCBI RefSeq database (https://www.ncbi.nlm.nih.gov/refseq/, accessed on 1 May 2022), including 2bRAD-M of bacteria, fungi, and archaea. The following formula was used to calculate the Gscore value of species [21]:$${\mathrm{Gscore}}_{\mathrm{soecies}\;i}=\sqrt{S_{i}\times {\;t}_{i}}$$

S: The number of reads mapped to all 2bRAD markers of species i in the sample; t: the number of species mapped to all 2bRAD markers of species i in the sample.

Species with a Gscore higher than 5 were selected as candidate species. The annotated information was obtained based on the microorganisms in each sample, the reads of each sample were counted based on the database, and the relative abundance of each microorganism in the sample was calculated for classification and clustering using the operational taxonomic unit (OTU) [20]. The alpha diversity and beta diversity analyses were conducted using the R package vegan 2.6.4 software package and photoseq 1.42.0 software package, and the diversity index of the different samples was calculated. The Kruskal–Wallis algorithm in R language was used for differential species analysis, and R language’s ggplot2 3.3.6, ggpubr 0.6.0, and pheatmap 1.0.12 software package was used to complete the drawing.

## 5. Conclusions

In this study, a novel *V. parahaemolyticus* bacteriophage vB_VpaP_SJSY21 was successfully isolated and characterized. A preliminary exploration of the effects of introducing phage SJSY21 into an aquaculture system on the gut microbiota of *L. vannamei* was conducted. These findings provide evidence of the potential of phage SJSY21 as an effective tool for preventing AHPND in shrimp farming. Furthermore, they provide new perspectives and resources for the application of bacteriophage therapy in the aquaculture industry in the future.

## Figures and Tables

**Figure 1 ijms-24-17529-f001:**
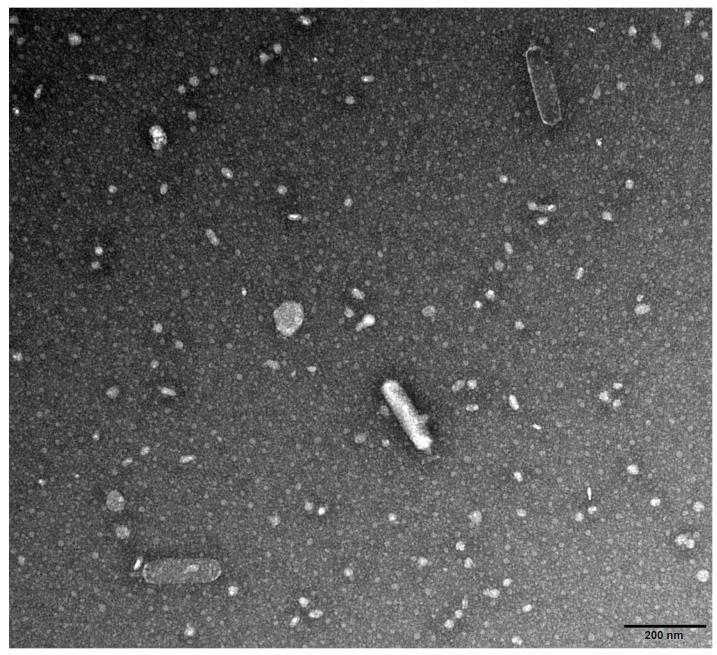
Transmission electron micrographs of bacteriophage vB_VpaP_SJSY21. The scale bar is in the lower right corner of the picture.

**Figure 2 ijms-24-17529-f002:**
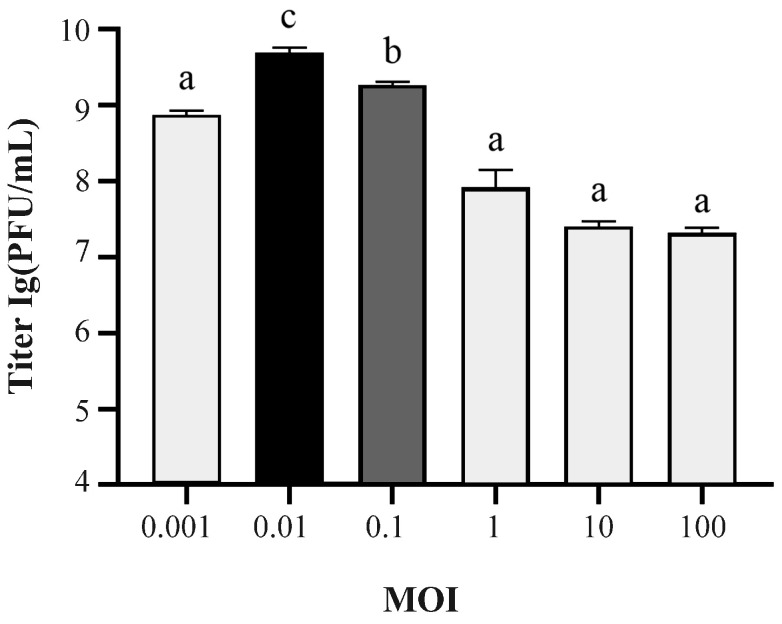
The titer of bacteriophage vB_VpaP_SJSY21 under different MOIs. Each column represents the mean of three independent replicates, error bars indicate the standard deviation, and the different letters represent significant differences (*p* < 0.05).

**Figure 3 ijms-24-17529-f003:**
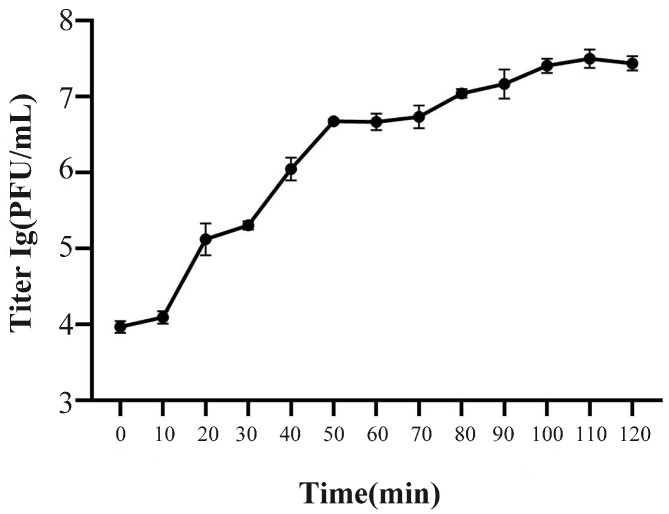
One-step growth curve of bacteriophage vB_VpaP_SJSY21. Each point represents the mean of three independent replicates and error bars indicate the standard deviation.

**Figure 4 ijms-24-17529-f004:**
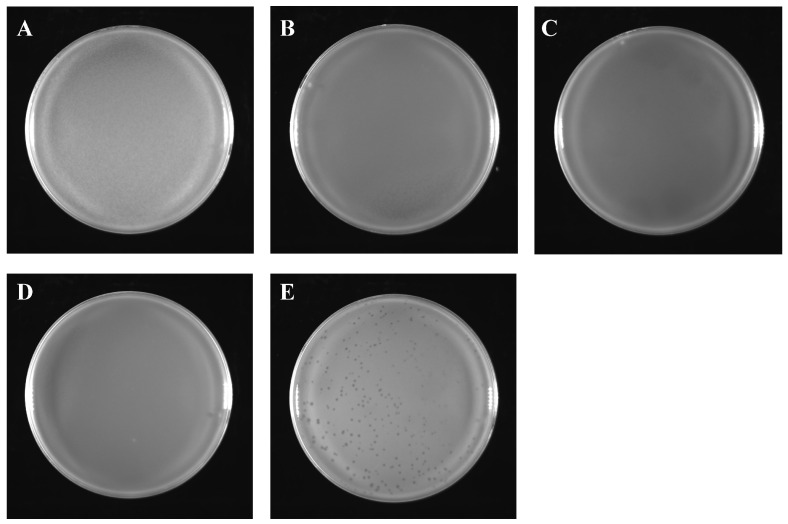
Lysis assay result of vB_VpaP_SJSY21. (**A**) *Clostridium butyricum*; (**B**) *Bacillus subtilis* strain A; (**C**) *Bacillus subtilis* strain B; (**D**) *Bacillus licheniformis*; (**E**) *Vibrio parahaemolyticus*.

**Figure 5 ijms-24-17529-f005:**
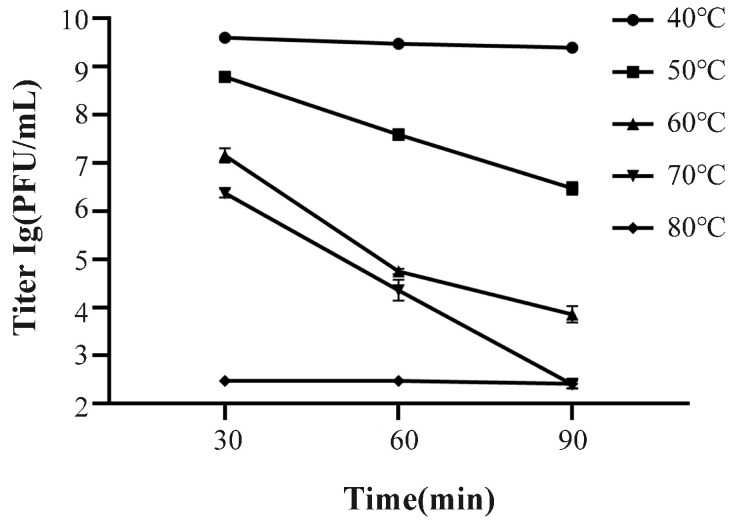
Thermal stability of bacteriophage vB_VpaP_SJSY21. Each point represents the mean of three independent replicates and error bars indicate the standard deviation.

**Figure 6 ijms-24-17529-f006:**
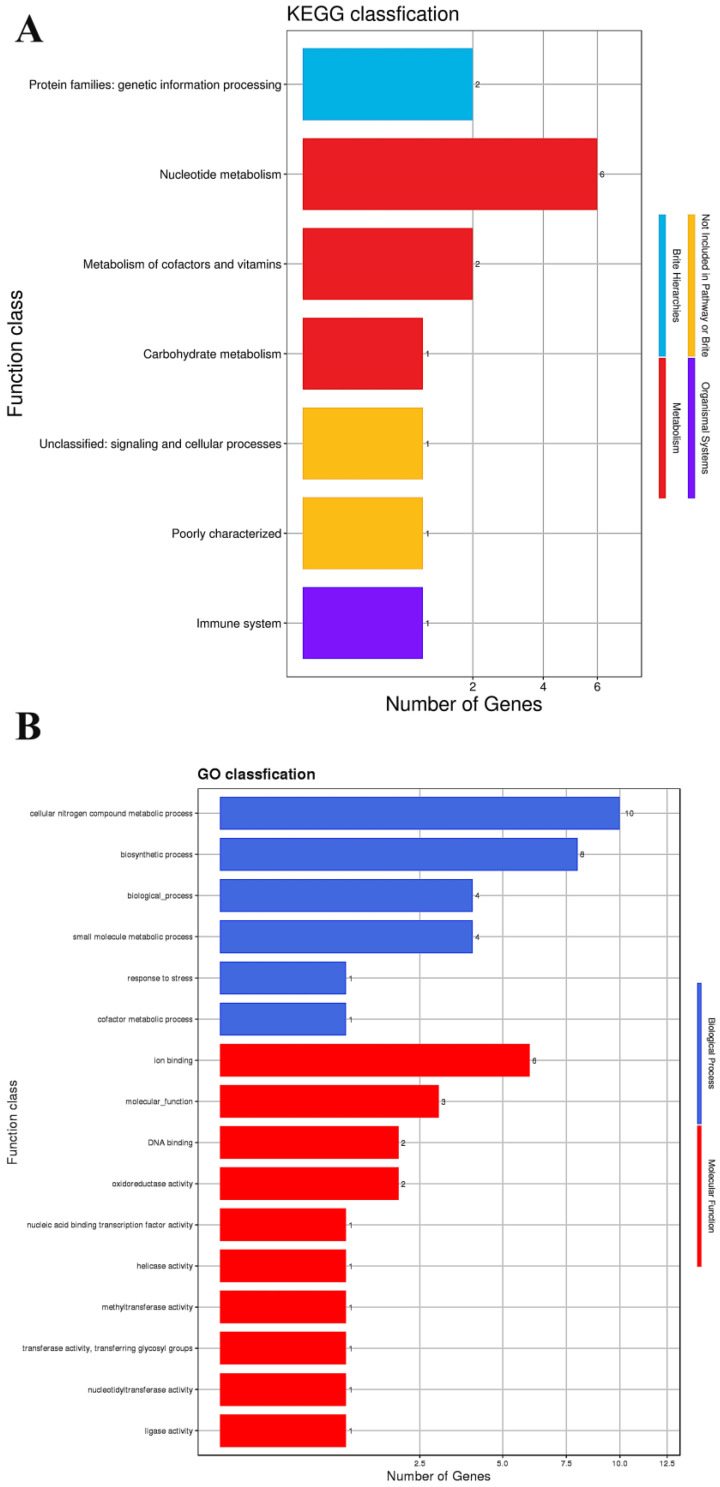
(**A**) KEGG pathway annotation result and (**B**) GO enrichment results of bacteriophage vB_VpaP_SJSY21 genome sequence.

**Figure 7 ijms-24-17529-f007:**
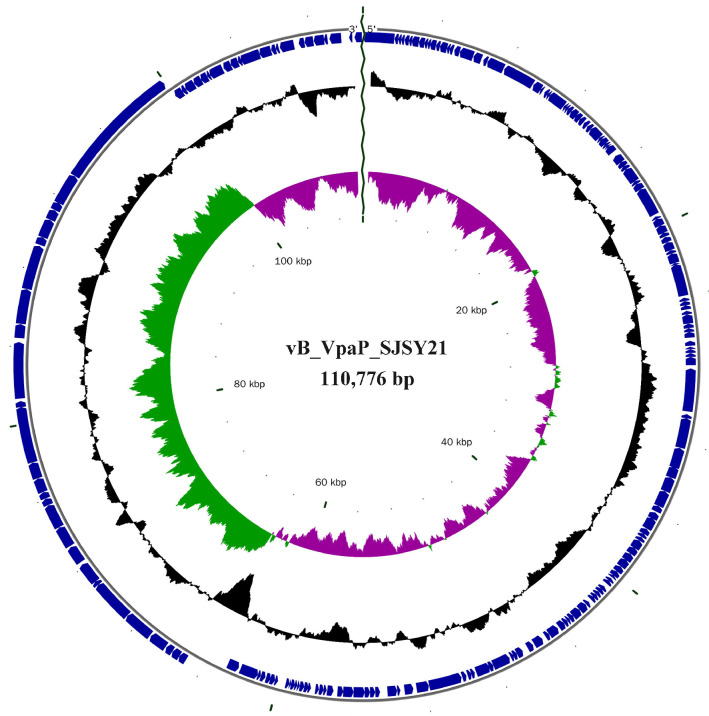
The genome map of vB_VpaP_SJSY21. From the inside to the outside, the first circle represents the scale, the second represents the GC Skew value, the third represents the GC content, and the fourth and fifth represent the location of CDS, tRNA, and rRNA on the genome.

**Figure 8 ijms-24-17529-f008:**
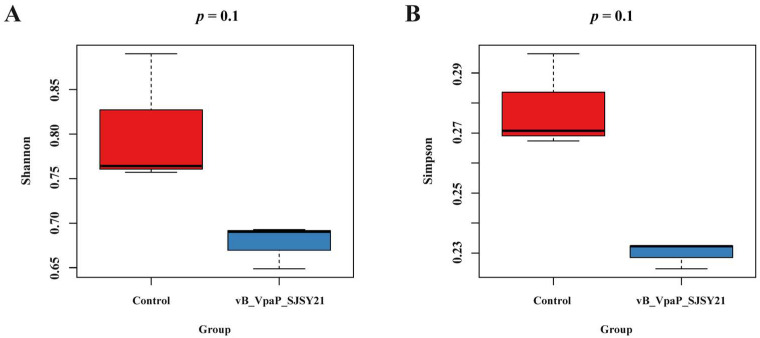
Shannon index (**A**) and Simpson index (**B**) of vB_VpaP_SJSY21 group and control group.

**Figure 9 ijms-24-17529-f009:**
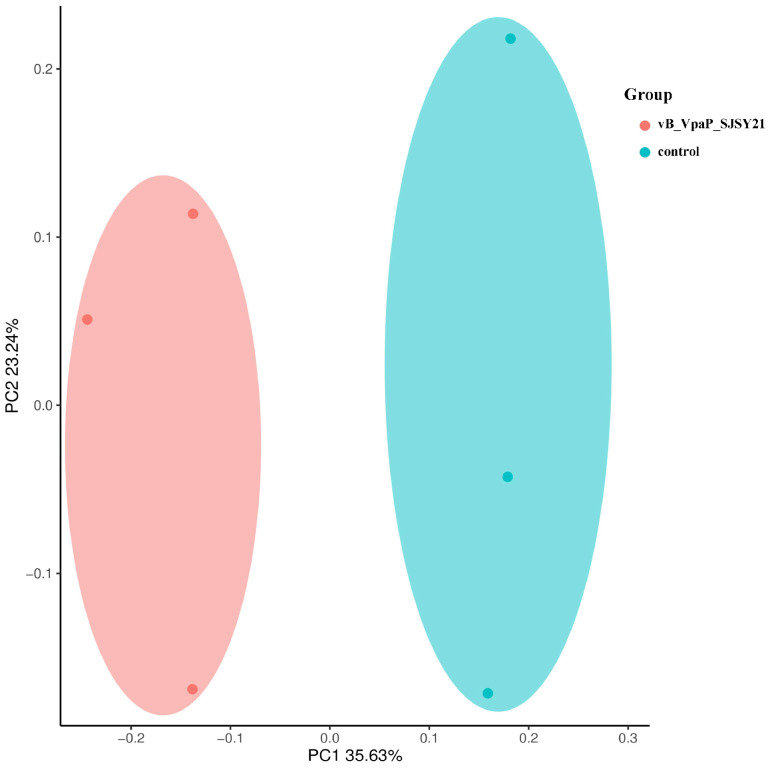
Scatter chart of intestinal microflora diversity of vB_VpaP_SJSY21 group and control group based on PCoA.

**Figure 10 ijms-24-17529-f010:**
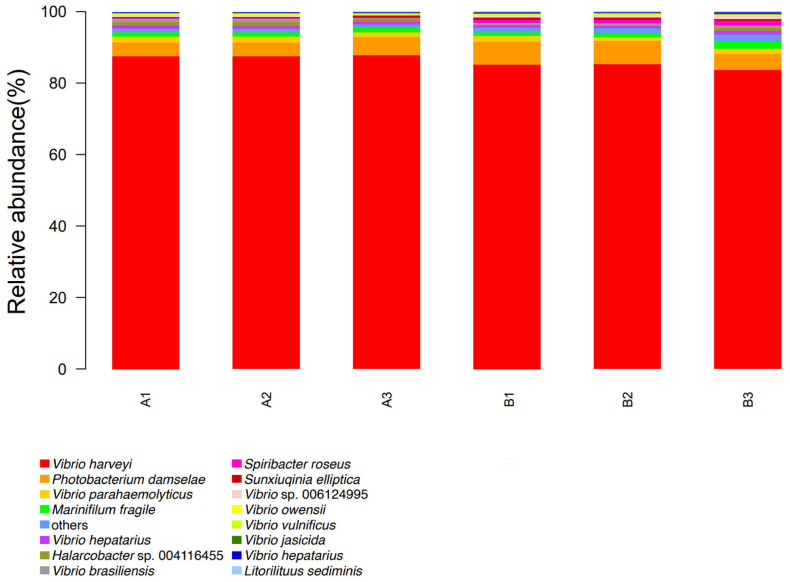
The composition structure of the top 15 bacterial groups in the relative abundance ranking of each sample at the species level. A: vB_VpaP_SJSY21 group; B: control group; others represent species outside the top 15 in relative abundance ranking.

**Figure 11 ijms-24-17529-f011:**
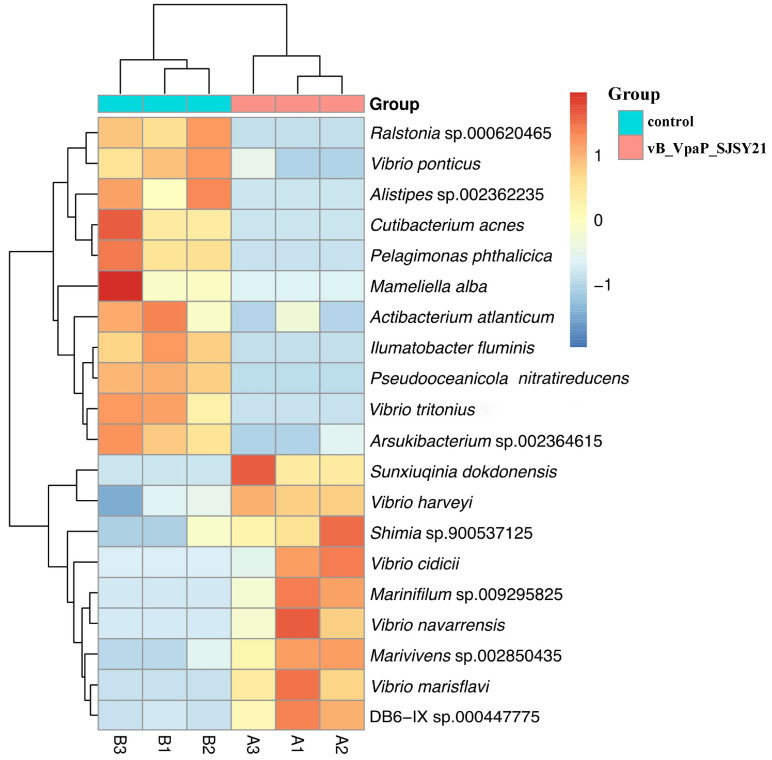
Analysis heatmap of different species. Each row represents a microorganism and each column represents a sample. The color of each grid in the heatmap represents the abundance of corresponding microbial species in the corresponding sample. High abundance is displayed in red, while low abundance is displayed in sky blue.

**Table 1 ijms-24-17529-t001:** Overview of vB_VpaP_SJSY21 whole-genome sequencing data filtering.

Read Number	Total Bases (bp)	N Rate (%)	GC Content (%)	Q20 Rate (%)	Q30 Rate (%)
20,297,674	3,044,540,950	0.000779	40.8	98.52	94.98

**Table 2 ijms-24-17529-t002:** Statistics of assembly results of whole-genome sequencing data of vB_VpaP_SJSY21.

Size (bp)	GC Content (%)	A (%)	G (%)	C (%)	T (%)
110,776	40.91	29.48	19.37	21.54	29.61

**Table 3 ijms-24-17529-t003:** EggNOG annotation of vB_VpaP_SJSY21 genome.

ORF Number	EggNOG Number	Prediction of Protein Functions
ORF 1	ENOG411EP44	DNA polymerase
ORF 17	ENOG411EPEK	PhoH-like protein
ORF 20	ENOG411EP1T	Deoxyribonucleoside diphosphate metabolic process
ORF 21	ENOG411EPH2	Ribonucleotide reductase, barrel domain
ORF 35	ENOG411EPBV	Flavin adenine dinucleotide binding
ORF 47	ENOG411EPAI	NAD biosynthetic process
ORF 53	ENOG411EP4J	Macro domain
ORF 55	ENOG411EPQT	Magnesium ion binding
ORF 57	ENOG411EP64	DNA helicase
ORF 71	ENOG411EP4Y	-
ORF 73	ENOG411EPJD	Glutamine amidotransferase domain
ORF 76	ENOG411EPAU	Phage phiEco32-like COOH.NH2 ligase-type 2
ORF 152	ENOG411EP64	DNA helicase
ORF 153	ENOG411EPA3	-
ORF 156	ENOG411EPGN	-
ORF 167	ENOG411EP2N	-
ORF 175	ENOG411EP0H	T4 recombination endonuclease VII
ORF 185	ENOG411EP9Z	Exonuclease activity
ORF 194	ENOG411EP4C	-

## Data Availability

2bRAD-M sequencing data from this study have been deposited in the GenBank Sequence Read Archive under accession number PRJNA1025901. The accession number of sequence data for *V. parahaemolyticus* bacteriophage vB_VpaP_SJSY21 in GenBank is OR723929. All detailed data presented in this study are available on request from the corresponding authors.

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
