# Peer review of "Characterization and Preliminary Application of a Novel Lytic Vibrio parahaemolyticus Bacteriophage vB_VpaP_SJSY21"

_ijms, 2023, doi:10.3390/ijms242417529_

Round 1

Reviewer 1 Report

Comments and Suggestions for Authors

This is a significant study in an important matter: the characterization and preliminary application of the Bacteriophage vB_VpaP_SJSY21.

Briefly, Litopenaeus vannamei (one of the most economically significant aquatic species globally)have been subject to the emergence of acute hepatopancreatic necrosis disease (AHPND) resulting in substantial losses within L. vannamei farming industry. Phage therapy can be an effective strategy for preventing and controlling bacterial infections like AHPND, thereby promoting the healthy and sustainable growth of the shrimp aquaculture sector. This work is a preliminary characterization and application study of this approach.

The work is well outlined and well organized. The experiments are well organized and the results support the conclusion. Overall, it is a significant study on this matter. However, in order to be published, in my opinion, there are some changes to be made:

Page 2; Line 70: Is it possible to have the phage image with higher magnification? It would be good to have it.

Page 3: Line 87: Please explain the sentence “The result of the lysis assay was shown in Figure 4 and phage SJSY21 did not interact with four probiotics selected in this study. As shown in Figure 5, phage SJSY21 maintained high activity at 40 â—¦C for 90 minutes, while at 50 â—¦C and above, but the titer of this bacteriophage decreased within 90 minutes. It is not clear or understandable.

Page 6, Figure 6: Image quality must be improved. It is not possible to clearly see it.

Page 7, Line 137: Figure 7 must be clearly explained.

Page 10; Line 211: Please comment on the disadvantages of decreased microbiota.

Typos:

Page 2; Lin 66: “vB_VpaP_SJSY21.The” must be “vB_VpaP_SJSY21. The”

Page 10; Line 191: “This observation underscores thesafety of phage” must be “This observation underscores the safety of phage”.

After these changes, the manuscript is able to be publishable.

Author Response

Point 1: Page 2; Line 70: Is it possible to have the phage image with higher magnification? It would be good to have it.

Response 1: Thank you for your encouragement and comments. But unfortunately, we are currently unable to obtain phage images with higher magnification.

Point 2: Page 3: Line 87: Please explain the sentence “The result of the lysis assay was shown in Figure 4 and phage SJSY21 did not interact with four probiotics selected in this study. As shown in Figure 5, phage SJSY21 maintained high activity at 40 â—¦C for 90 minutes, while at 50 â—¦C and above, but the titer of this bacteriophage decreased within 90 minutes. It is not clear or understandable.

Response 2: The explanation of Figure 4 has been added to the latest manuscript

Point 3: Page 6, Figure 6: Image quality must be improved. It is not possible to clearly see it.

Response 3: The higher quality Figure 6 has been replaced in the latest manuscript.

Point 4: Page 7, Line 137: Figure 8 must be clearly explained.

Response 4: Figure 8 has been explained in the latest manuscript.

Point 5: Page 10; Line 211: Please comment on the disadvantages of decreased microbiota.

Response 5: Comments on the disadvantages of microbial community reduction have been added to the latest manuscript.

Point 6: Page 2; Lin 66: “vB_VpaP_SJSY21.The” must be “vB_VpaP_SJSY21. The”

Response 6: It has been corrected as “vB_VpaP_SJSY21. The”.

Point 7: Page 10; Line 191: “This observation underscores thesafety of phage” must be “This observation underscores the safety of phage”.

Response 7: It has been corrected as “This observation underscores the safety of phage”.

Reviewer 2 Report

Comments and Suggestions for Authors

Dear Authors

The manuscript submitted for evaluation covers the development of alternative methods to antibiotics to control the Vibrio parahaemolyticus bacteria; Litopenaeus vannamei in aquaculture.

The research seems to be interesting and contributes many important elements to the scientific discipline presented by the authors.

However, the authors did not explain why these studies were carried out at all and what their potential impact on the development of a given scientific discipline will be. In many places, especially in the "material and methods" chapter, the authors do not cite the literature regarding the presented methodology and formulas used in the calculation.

It is advisable to redact the entire text in the selected fragments in accordance with the suggestions contained in the manuscript.

In my opinion, the manuscript requires corrections to be accepted for publication.

​

Author Response

Point 1: Page 1, Line 39-43, in this part of the chapter the Authors should add some information about the prevalence of these infections. Some characteristocs like antibiotic resistance in boths pathoges. Is it a global problem? Why it's necessary to implement the bacteriophages?

Response 1:  Thank you for your suggestions and comments. The relevant information and references have been added to the latest version of the manuscript.

Point 2: Page 2, Line 54-60, this part of the texts should be rewritten. The Authors should remove the information about the isolation of phages buy should add important information why They do to carry out this study. What was the mean reason of this study? And what was the influence on the development of this part of the sciences?

Response 2: Thanks for your suggestion. This section of the text has been rewritten, and information on phage isolation has been deleted, emphasizing the main reasons for conducting this research and its impact on the development of phage science.

Point 3: Page 11, Line 259-262, Please add some references

Response 3: The relevant references has been added to the latest manuscript

Point 4: Page 11, Line 266-269, Please add some references

Response 4: The relevant references has been added to the latest manuscript

Point 5: Page 11, Line 281-283, Please add some references

Response 5: The relevant references has been added to the latest manuscript

Point 6: Page 12, Line 313, Please add more information about the experimental animals, the conditions, age or weight? Total number?

Response 6: The information on experimental animals has been added to the latest manuscript.

Point 7: Page 13, Line 340-341, Please add some references

Response 7: The relevant references has been added to the latest manuscript.